# Association between arsenic exposure and intrauterine growth restriction: A systematic review and meta-analysis

Jing Jiang[1,2☯], Xuan Zuo[1,2☯], Songlin An[1,3], Jing Yang[1,2], Linfei Wu[1,2], Rong Zeng[1,2], Qiongdan Hu[1,2], Lu Fan[1,2], Haiyu Wang[1,2], Chuanwu Yang[1,2], Yihan Liang[1,2], Yuanzhong Zhou[1,2], Hong Pan[1,2*], Yan Xie[1,2*]

**1** School of Public Health, Zunyi Medical University, Zunyi, China, **2** Key Laboratory of Maternal & Child Health and Exposure Science of Guizhou Higher Education Institutes, Zunyi, China, **3** Department of Stomatology, Daping Hospital, Army Medical University (The Third Military Medical University), Chongqing, China

☯ Equal contribution: These authors contributed equally to this work and should be considered co-first authors.

* ph-tian999@163.com (HP); xie814yan@zmu.edu.cn (YX)

## Abstract

Several observational studies have explored the link between arsenic (As) exposure and intrauterine growth restriction (IUGR). However, epidemiological findings have been inconsistent, with a wide range of reported heterogeneity. This study aims to systematically evaluate the association between As exposure and IUGR (SGA(Small for gestational age), PTB(Preterm birth), LBW(Preterm birth)) through a meta-analysis. We searched six databases—China National Knowledge Infrastructure, Wan Fang, VIP Database, PubMed, Web of Science, and Science Direct—for studies on As exposure and IUGR up to May 2024. After screening and data extraction, a comprehensive bias risk assessment was conducted using the Newcastle-Ottawa Scale (NOS), AHRQ (the assessment tool of the Agency for Healthcare Research and Quality), and NTP/OHAT (the assessment tool of the National Toxicology Program/Office of Health Assessment and Translation). Meta-analysis was conducted using random-effects models ($I^2 > 50\%$) or fixed-effects models ($I^2 < 50\%$) to estimate effect sizes. Subgroup analysis and meta-regression analysis were performed to identify the sources of heterogeneity. Publication bias was assessed using the Egger test, Begg test, and funnel plot. Eleven studies, including 2,183,652 participants from the Americas, Europe, Asia, and Africa, were analyzed. Results showed a significant association between As exposure and SGA (OR: 1.06, 95% CI: 1.00, 1.13), particularly in Asia (OR: 1.28, 95% CI: 1.10, 1.49). Maternal exposure to higher As levels (10–100 µg/L) was also significantly associated with SGA (OR: 1.25, 95% CI: 1.04, 1.50). Although PTB (OR: 1.03, 95% CI: 0.99, 1.07) and LBW (OR: 1.03, 95% CI: 0.97, 1.09) did not show overall significant associations, subgroup analyses revealed increased risks under specific conditions. As exposure at 1–10 µg/L significantly increased PTB risk (OR: 1.13, 95% CI: 1.06, 1.21), while exposure at 0–1 µg/L significantly increased LBW risk (OR: 1.13, 95% CI: 1.06, 1.21). This study supports a link between As exposure and increased IUGR risk, particularly SGA. Stricter public health policies are needed to reduce arsenic exposure during

**Data availability statement:** All relevant data are within the manuscript and its Supporting Information files.

**Funding:** This work was supported by Nature Science Foundation of Guizhou Province (QKH-J[2022]YB612),the City School Joint Foundation Project (QKH-PTRC[2020]-018 & ZYKH-HZ-Z[2021]292), City School Joint Foundation Project (QKH-PTRC[2019]-003 & ZYKH-HZ-Z[2020]64), Start-up Foundation for Doctors of Zunyi Medical University (QKH-PTRC[2019]-032), Guizhou Provincial Education Reform Project (SJJG2022-02-166), and the High-level innovative talents in Guizhou Province (GCC[2022]039-1), Postgraduate Research Fund project of Zunyi Medical University (ZYK216), Postgraduate Research Fund project of Zunyi Medical University (ZYK214),Scientific Research Program of Guizhou Provincial Department of Education (QJJ [2023] 019).

**Competing interests:** The authors have declared that no competing interests exist.

**Abbreviations:** As = Arsenic, AHRQ = Agency for Healthcare Research and Quality, CI = Confidence interval, LBW = Low birth weight, NOS = Newcastle–Ottawa scale, NTP/OHAT = National Toxicology Program/Office of Health Assessment and Translation, OR = Odds ratio, PTB = Preterm birth, SGA = Small for gestational age

pregnancy. However, due to heterogeneity and potential publication bias, results should be interpreted with caution.

## Introduction

Intrauterine growth restriction (IUGR) is a common and severe fetal developmental disorder, with a global incidence rate of approximately 3% to 7% [1]. IUGR not only significantly increases the risk of perinatal mortality but is also closely associated with neurodevelopmental delays in the neonatal period and chronic diseases such as cardiovascular disease and metabolic syndrome in adulthood, posing a serious threat to long-term health and quality of life [2,3]. Typical manifestations of IUGR include low birth weight (LBW), small for gestational age (SGA), and preterm birth (PTB) [4–7]. These adverse pregnancy outcomes represent significant public health challenges globally, particularly in resource-limited settings [8].

The etiology of IUGR is complex and multifactorial, involving maternal, placental, and environmental factors. Maternal factors such as malnutrition, gestational hypertension, and uterine structural abnormalities play significant roles in the development of IUGR. At the same time, placental dysfunction is widely recognized as one of the critical mechanisms leading to IUGR [9]. Studies have shown that the birth weight of IUGR infants is often below the 10th percentile for their gestational age, which significantly increases the incidence of SGA [10]. Another study found that the risk of LBW is significantly higher in IUGR infants compared to non-IUGR infants, closely related to insufficient nutrient supply caused by placental insufficiency [11]. Furthermore, IUGR significantly increases the risk of preterm birth (PTB), particularly when accompanied by placental dysfunction, leading to a substantial rise in PTB rates [12].

In recent years, environmental toxins, particularly arsenic (As), have garnered increasing attention in IUGR research. As is a potent toxin widely present in the environment and is classified as a Group 1 carcinogen by the International Agency for Research on Cancer (IARC) [13]. The primary exposure pathways for As include drinking water, food, air, skin contact, and maternal-fetal transmission [14–18]. Although the World Health Organization (WHO) recommends that As concentrations in drinking water be below 10 μg/L [19], studies have shown that even at this level, As exposure can significantly increase the risk of various cancers, including kidney and prostate cancer [20]. Additionally, As exposure is closely associated with the occurrence of chronic diseases such as cardiovascular disease, diabetes, and neurological damage [21]. During pregnancy, As can cross the placental barrier, leading to irreversible damage to fetal development, thereby further increasing the risk of IUGR [22].

Although epidemiological studies have explored the association between prenatal As exposure and IUGR, the results have been inconsistent. Some studies have found that even low-dose As exposure can significantly increase the risk of IUGR [23], while others have not observed a significant association [24]. These inconsistencies may result from differences in study design, exposure assessment, exposure levels, and study populations. Moreover, animal models and in vitro cell experiments also support the adverse effects of As exposure on fetal development. As exposure has been shown to disrupt placental function by inducing oxidative stress and DNA damage, ultimately leading to restricted fetal nutrient absorption [25–27].

In summary, the existing evidence is inconsistent, but given the widespread global prevalence of As exposure and its potential harm to maternal and fetal health, we conducted a systematic review and meta-analysis to evaluate the association between As exposure and IUGR. The findings aim to provide a scientific basis for optimizing pregnancy health management strategies and informing public health policy decisions.

## Materials and methods

The study protocol was registered with Prospero in compliance with PRISMA guidelines [28] (CRD42023473902, https://www.crd.york.ac.uk/PROSPERO/). A completed PRISMA-P checklist for the current review is provided in S1 Table.

### Literature search

We determined various digital repositories to collect information on the effects of As exposure on pregnancy outcomes. We searched the PubMed, Web of Science, and Science Direct databases and collected all the relevant English-language research published up to May 2024. We followed a similar approach for accessing, gathering literature, and referencing the Chinese databases (China National Knowledge Infrastructure Wanfang and VIP database) as we did for English-language research literature databases, including manual retrieval of research citations. We used the following strategy for literature retrieval by PubMed: ("pregnancy" OR "pregnant woman" OR "gestation" OR "pregnancy outcome" OR "women pregnant") AND ("Arsenic" OR "arsenic oxide" OR "$As_2O_3$" OR "diarsenic trioxide" OR "arsenic compounds"). We used a combination of relevant words from medical topics and textual content in our searches. Two authors (JJ and XZ) independently reviewed the literature and selected qualified studies. Any discrepancies were resolved through dialog or consulting with a third evaluator (YX). For a comprehensive understanding of the methods used to access other digital databases, please refer to S2 Table.

### Inclusion and exclusion criteria

All selected studies met specific inclusion criteria: 1) The study participants (P) were pregnant women; 2) The studies (S) were population-based observational studies, such as case-control studies, cohort studies, and cross-sectional studies; 3) The studies provided risk assessments and 95% confidence intervals (CI) linking As exposure (E) to IUGR (O), or provided sufficient data for basic risk calculations and variance analysis. We excluded the following studies: 1) Certain types of medical publications, such as reviews, meta-analyses, or conference abstracts; 2) Studies involving animals or molecular experiments; 3) Studies with data overlap with other publications or without extractable data.

### Study quality assessment

Due to their inclusive characteristics, the included studies are observational in design. To assess the methodological robustness of cohort studies, the researchers used the Newcastle-Ottawa Scale (NOS) [29]. For cross-sectional studies, the researchers followed the evaluation criteria set by the Agency for Healthcare Research and Quality (AHRQ) [30]. Articles with a score >6 were considered high quality. Additionally, we further evaluated the quality of the included studies using the NTP/OHAT risk of bias rating tool for human and animal studies. This tool assesses each study based on seven risk of bias questions; detailed information on these questions and the assessment criteria can be found in S3 Table, "NTP/OHAT Risk of Bias Rating Tool." The initial assessment of the studies was conducted independently by two reviewers (JJ and XZ). After completing the assessments, the results from both reviewers were compared. In cases of disagreement, a third reviewer, YX, was invited to participate in the discussion or arbitration process.

### Screening and data extraction

This study conducted a literature search across multiple databases, including CNKI, Wanfang, VIP, PubMed, Web of Science, and ScienceDirect, identifying a total of 12,066 records. After

removing duplicates, 10,833 records were further screened based on their titles and abstracts (S4 Table), of which 10,766 were excluded as they did not meet the inclusion criteria. In the subsequent screening process, 67 articles were assessed for eligibility after full-text review, including 9 Chinese-language articles and 58 English-language articles. Ultimately, 56 articles were excluded due to irrelevant risk factors, data, or outcomes: 2 articles had no relevant risk factors, 36 had no relevant data, 12 had no relevant outcomes, and 6 had no alignment with the research objectives. The remaining 11 articles were included in the systematic review and meta-analysis.

The following details were provided for each article: the name of the primary author, region, study design, date, sample size, study outcomes, techniques used to measure the type of As, exposure duration, publication date, and risk estimates (odds ratio [OR] and relative risk [RR]). Data extraction was conducted by two independent reviewers (JJ and XZ). After extraction, the results were compared to ensure accuracy. In cases of disagreement, the third reviewer, YX, was consulted for discussion or arbitration.

## Statistical analyses

Data extraction, compilation, and summarization were performed using Excel 2022. Meta-analysis was conducted using Stata 16.0 (STATA, College Station, Texas, USA). Given the differences in study design, sample types, and regions among the included studies, as well as the potential for heterogeneity, heterogeneity among the studies was assessed using the $I^2$ statistic. When the heterogeneity test yielded a P-value > 0.05 and an $I^2$ value < 50%, a fixed-effects model was used for the meta-analysis. Conversely, if the P-value was ≤ 0.05 or the $I^2$ value was ≥ 50%, indicating significant heterogeneity, a random-effects model was applied [31]. Subgroup analysis and meta-regression analysis were conducted to identify potential sources of heterogeneity. Additionally, sensitivity analysis was performed by sequentially excluding studies to verify the robustness of the results. Publication bias was assessed using the Egger test and Begg test, and the results were visualized using a funnel plot.

## Results

### Literature selection and study characteristics

In the initial search, we identified a total of 12,066 articles from three English electronic databases (PubMed, Web of Science, and Science Direct) and three Chinese electronic databases (China National Knowledge Infrastructure, Wan Fang, and VIP Database). After reviewing titles, and abstracts, 10,766 studies were excluded, and 67 full-text articles were selected for further analysis. Among these 67 articles, 56 were excluded after full-text review (S5 Table), including 2 articles that did not address the risk factors of interest, 36 that lacked relevant data, 12 that did not report the outcomes of interest, and 6 that did not clearly define key study objectives. Ultimately, 11 articles met the inclusion and exclusion criteria and were included in this study (S6 Table). The flowchart of the meta-analysis screening process is shown in Fig 1. These studies comprised 10 cohort studies [32–41]and 1 cross-sectional study [42]. The studies were conducted in five different regions: the Americas (n = 4), Europe (n = 2), Southeast Asia (n = 2), Asia (n = 2), and Africa (n = 1) (S7 Table). Among the 11 studies considered, 9 examined the effects of As exposure during pregnancy on SGA, 5 studies investigated the impact of As exposure during pregnancy on LBW, and 8 studies reported on the effects of As exposure during pregnancy on PTB. The biological samples collected included urine, drinking water, blood, placenta, and moss samples. The methods used to measure As exposure in the reviewed studies primarily included techniques such as inductively coupled plasma mass spectrometry (ICP-MS) and hydride generation atomic fluorescence spectrometry (HG-AFS) (Table 1).

| | Selection Bias | Confounding Bias | Attrition/Exclusion Bias | Detection bias: Exposure Characterization | Detection Bias: Outcome Characterization | Selective Reporting Bias | Conflict of interest |
|---|---|---|---|---|---|---|---|
| Thomas et al. 2015 | + | + | + | ++ | + | ++ | ++ |
| Bloom et al. 2016 | + | + | + | ++ | + | ++ | ++ |
| Almberg et al. 2017 | + | + | + | + | + | + | ++ |
| Wai et al. 2017 | + | + | + | ++ | + | ++ | ++ |
| Liu et al. 2018 | + | + | + | ++ | + | ++ | ++ |
| Wang et al. 2018 | + | + | + | ++ | + | + | ++ |
| Freire et al. 2019 | + | + | - | ++ | + | ++ | ++ |
| Mullin et al. 2019 | + | + | + | + | + | ++ | ++ |
| Nyanza et al. 2020 | + | + | - | ++ | + | ++ | ++ |
| Comess et al. 2021 | + | + | + | + | ++ | ++ | ++ |
| Fano-Sizgorich et al. 2021 | + | + | + | ++ | + | ++ | ++ |

**Fig 1. PRISMA flowchart of the searching and selecting process.**

## Quality assessment of the studies

The quality of the included studies was assessed using the Newcastle-Ottawa Scale (NOS) and the evaluation criteria set by the Agency for Healthcare Research and Quality (AHRQ). The cross-sectional study scored 7 points, and the average score for all cohort studies was 7.818 (≥6) (S8 Table), indicating that the quality of the included studies was relatively high. Additionally, we further evaluated the quality of the included studies using the NTP/OHAT risk of bias assessment tool for human and animal studies. The results showed that the "selection bias" in the included studies was almost universally rated as "probably low" because the inclusion and exclusion methods for participants were consistent across the studies. "Confounding bias" was also generally rated as "probably low" since the included studies considered and controlled for known confounders (e.g., maternal age, pre-pregnancy BMI, socioeconomic status, smoking status, parity, etc.). For "attrition/exclusion bias," studies were rated as "probably high risk" if they did not address or adequately explain data attrition/exclusion, and "probably low risk" if there was minimal attrition/exclusion with clear data handling. Three studies were rated as "probably low risk" for "detection bias: exposure characteristics," while studies that used direct measurement methods with stringent quality control (e.g., direct measurement of metal concentrations in blood or urine) were rated as "low risk." Nearly all studies used standard clinical methods to assess birth outcomes, so "detection bias: outcome characterization" was rated as "probably low risk." Nine studies fully reported the pre-specified outcomes and conducted sensitivity analyses, leading to an assessment of "low risk" for "selective reporting bias," while two studies with limited discussion of secondary outcomes were rated as "probably low risk." Almost all studies clearly described "funding support" and "no conflict of interest," so "conflict of interest" was rated as "low risk" in all 11 studies (Fig 2).

**Table 1. Study characteristics in meta-analysis.**

| Author (year) | Country | Study design | Partic-ipant | Exposure [As level (µg/L)] | Control [As level (µg/L)] | Outcome | Total cases (E/C) | Sample | Exposure Time | Testing method | Quality score |
|---|---|---|---|---|---|---|---|---|---|---|---|
| Thomas et al. (2015)[d,2] | Canada (North America) | CO | P | ≥0.525 | <0.525 | SGA | 1337/498 | Urine | Throughout the entire pregnancy | Inductively-coupled plasma mass spectrometry | 8 |
| Bloom et al. (2016)[f,2] | Romania (Europe) | CO | P | ≥10 | <10 | SGA,PTB | 10/122 | Drinking water | Throughout the entire pregnancy | Hydride gen-eration atomic absorption spectrometry | 8 |
| Almberg et al. (2017)[a,2] | USA (North America) | CO | P | ≥10 | <10 | SGA, LBW, VLBW, PTB, VPTB | 289091/139713,260060 /168744,289091/13971 3,289303/139501,2893 03/139501 | Drinking water | Throughout the entire pregnancy | Not available | 7 |
| Wai et al. (2017)[g,1] | Myanmar (Southeast Asia) | CO | P | ≥74 | <74 | LBW,PTB | 26/393 | Urine | Throughout the entire pregnancy | Inductively-coupled plasma mass spectrometry | 9 |
| Liu et al. (2018)[f,2] | China (Asia) | CO | P | ≥20 | <20 | SGA | 151/1239 | Urine | Throughout the entire pregnancy | Inductively-coupled plasma mass spectrometry | 8 |
| Wang et al. (2018)[b,2] | China (Asia) | CO | P | ≥6.68 | <6.68 | LBW, SGA | 52/2344,183/2213 | Maternal serum | Early and mid-pregnancy | Hydride gen-eration atomic fluorescence spectrometry | 7 |
| Freire et al. (2019)[a,2] | Spain (Europe) | CO | P | ≥0.004 ng/g | <0.004 ng/g | LBW, SGA, PTB | 11/316,20/307,5/322 | Placentas | Throughout the entire pregnancy | Hydride gen-eration atomic fluorescence spectrometry | 8 |
| Mullin et al. (2019)[c,2] | Mexico (North America) | CO | P | ≥0.85 | <0.85 | SGA | 129/597 | Blood | Mid to late pregnancy | Inductively-coupled plasma mass spectrometry | 8 |
| Nyanza et al. (2020)[e,2] | Tanzania (Africa) | CO | P | ≥6.3 | <6.3 | PTB, LBW | 206/755,139/822 | Urine | Throughout the entire pregnancy | Inductively-coupled plasma mass spectrometry | 9 |
| Comess et al. (2021)[b,2] | USA (North America) | CO | P | ≥0.18 | <0.18 | PTB, VPTB, SGA | 3667/5138,508/ 8297,4630/4175 | Moss | Throughout the entire pregnancy | Inductively-coupled plasma-optical emission spectrophotometer | 7 |
| Fano-Sizgorich et al. (2021) | Peru (South America) | CS | P | ≥43.97 | <43.97 | SGA,PTB | 9/138 | Urine | Mid-pregnancy | Inductively-coupled plasma mass spectrometry | 7 |

Note: As, arsenic; SGA, small for gestational age; LBW, low birth weight; VLBW, very low birth weight; PTB, preterm birth; VPTB, very preterm birth; USA, United States of America; P, pregnant women; E, case group; C, control group; CO, cohort study; CS, cross-sectional study; Length of the study (years): a, 11; b, 9; c, 8; d, 7; e, 5; f, 4; g, 1; Exposure period: 1, third trimester; 2, entire pregnancy.

## Meta-analysis results

**Association between as exposure and SGA.** The meta-analysis revealed a significant association between As exposure and an increased risk of SGA. The overall analysis showed a combined odds ratio (OR: 1.06, 95% CI: 1.00, 1.13) (Fig 3), indicating that As exposure may be a risk factor for SGA. However, to identify potential sources of heterogeneity ($I^2 \geq 50\%$), a subgroup analysis was conducted. In the "Asia" subgroup, two studies indicated a significant positive correlation between As exposure and the risk of SGA (OR: 1.28, 95% CI: 1.10, 1.49), while no significant association was observed in studies conducted in the United States and

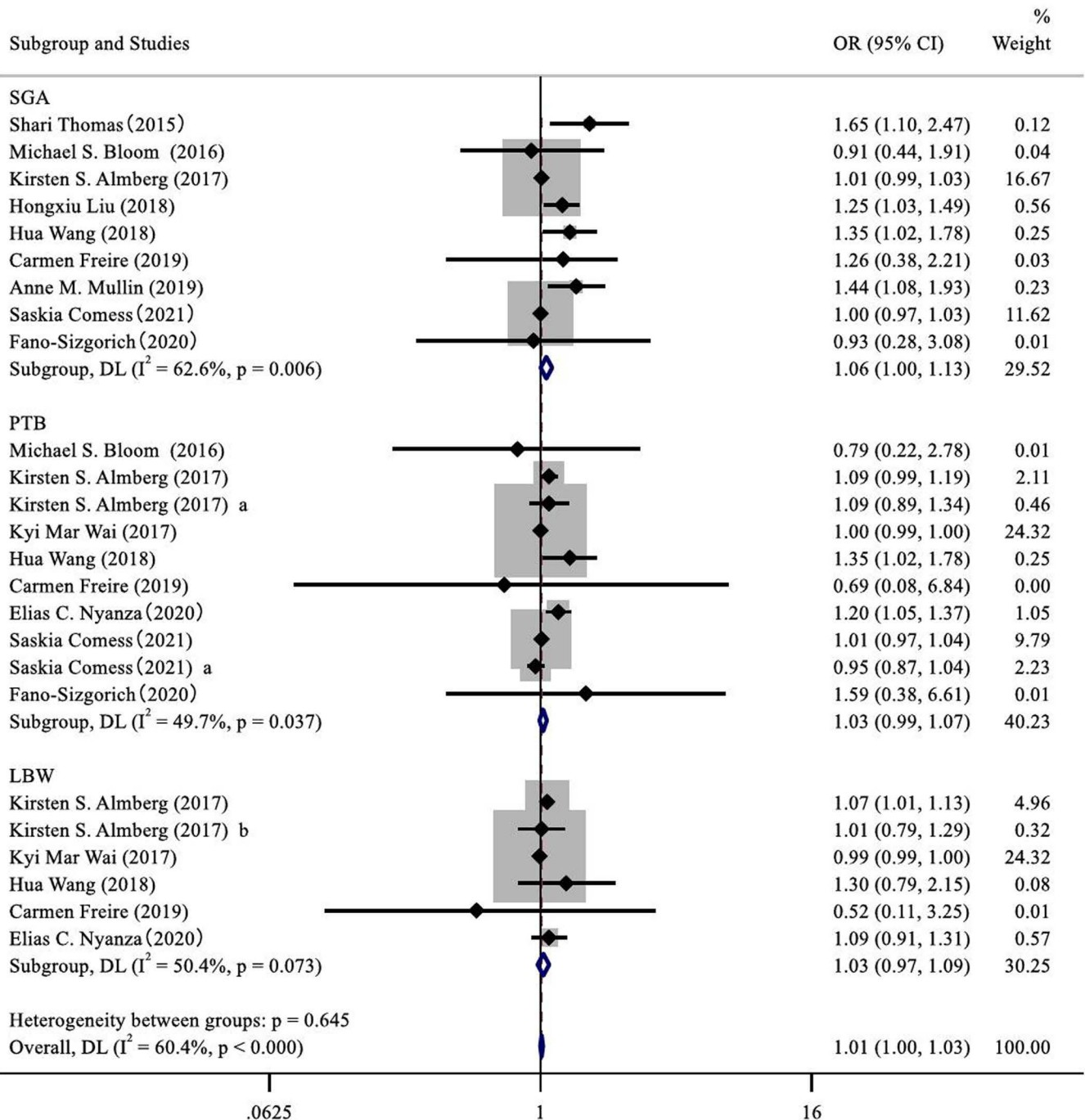

**Fig 2. Risk of bias summary using the OHAT framework.** ++ low, + probably low, − probably high, − − high.

Europe (Table 2). Additionally, in terms of sample type, a significant positive correlation was observed between SGA and As levels in blood (OR: 1.39, 95% CI: 1.14, 1.70) as well as As levels in urine (OR: 1.35, 95% CI: 1.06, 1.73). Furthermore, two studies using the HG-AFS method (OR: 1.34, 95% CI: 1.03, 1.75) and four studies using the ICP-MS method (OR: 1.25, 95%

CI: 1.00, 1.57) both indicated a correlation between As exposure and the occurrence of SGA. Subgroup analysis did not reveal a statistically significant association between SGA and lower levels of As exposure (0–1/1–10 μg/L) or As in drinking water samples. However, maternal exposure to higher levels of As ≥74 μg/L (10–100 μg/L) was significantly associated with the occurrence of SGA (OR: 1.25, 95% CI: 1.04, 1.50). This finding is consistent with the conclusions of the included studies, suggesting that As exposure may contribute to the occurrence of SGA.

**Association between as exposure and PTB.** The meta-analysis explored the correlation between As exposure and PTB. The results showed a combined odds ratio (OR: 1.03, 95% CI: 0.99, 1.07) (Fig 3), indicating no significant association between As exposure and PTB risk. The I² statistic was 49.7%. To further clarify the correlation between As exposure and PTB in the included studies, a subgroup analysis was conducted. However, the analysis found a significant positive correlation between As exposure and PTB risk in the "Africa" subgroup (OR: 1.20, 95% CI: 1.05, 1.37). A significant association was observed between As exposure and PTB when blood samples were used for detection (OR: 1.35, 95% CI: 1.02, 1.78). Studies using drinking water as the detection sample also showed a significant association between As exposure and PTB (OR: 1.09, 95% CI: 1.00, 1.18). Additionally, two studies that employed the HG-AFS technique indicated a significant association between As exposure and PTB (OR: 1.34, 95% CI: 1.01, 1.76). Among five studies, maternal As exposure at the 1–10 μg/L level was

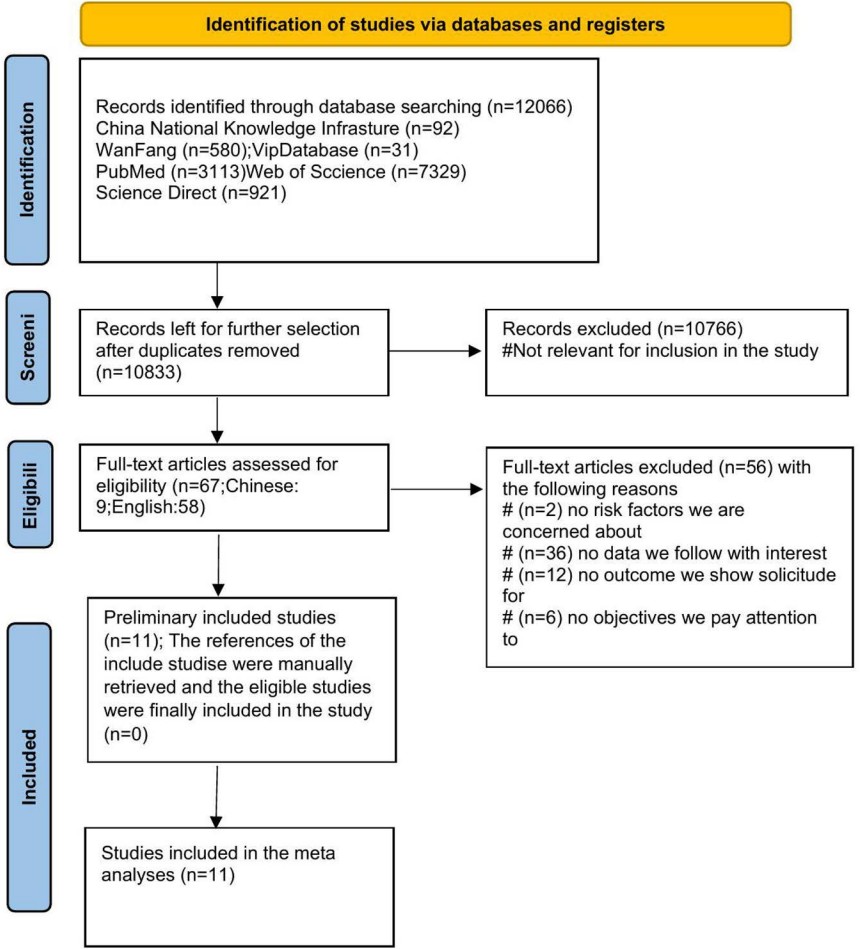

**Fig 3. Forest plot of meta-analysis on arsenic exposure and IUGR risk.**

**Table 2. Subgroup analysis of As exposure and IUGR risk.**

| Subgroup | Number of literature | Pooled estimates (95%CI) | I² (100%) | P |
|---|---|---|---|---|
| **SGA (As)** | | | | |
| *Region* | | | | |
| Africa | 0 | / | / | / |
| America | 4 | 1.03 (0.97,1.08) | 74.6 | 0.008 |
| Asia | 2 | **1.28 (1.10,1.49)** | 0.0 | 0.652 |
| Europe | 2 | 1.04 (0.59,1.83) | 0.0 | 0.578 |
| *Sample* | | | | |
| Blood | 2 | **1.39 (1.14,1.70)** | 0.0 | 0.753 |
| Urine | 2 | **1.35 (1.06,1.73)** | 33.2 | 0.221 |
| Drinking water | 2 | 1.01 (0.99,1.03) | 0.0 | 0.781 |
| *Testing method* | | | | |
| HG-AFS | 2 | **1.34 (1.03,1.75)** | 0.0 | 0.884 |
| ICP-MS | 4 | **1.25 (1.00,1.57)** | 82.3 | <0.000 |
| *As Exposure level* | | | | |
| 0–1 μg/l | 3 | 1.28 (0.91,1.80) | 83.0 | 0.003 |
| 1–10 μg/l | 3 | 1.10 (0.88,1.36) | 52.7 | 0.121 |
| 10–100 μg/l | 1 | **1.25 (1.04,1.50)** | 0.0 | <0.000 |
| **PTB (As)** | | | | |
| *Region* | | | | |
| Africa | 1 | **1.20(1.05,1.37)** | 0.0 | <0.000 |
| America | 5 | 1.02 (0.97,1.07) | 39.3 | 0.176 |
| Asia | 2 | 1.12 (0.84,1.50) | 77.6 | 0.035 |
| Europe | 2 | 0.76 (0.25,2.30) | 0.0 | 0.917 |
| *Sample* | | | | |
| Blood | 1 | **1.35 (1.02,1.78)** | 0.0 | <0.000 |
| Urine | 2 | 1.08 (0.91,1.29) | 86.1 | 0.007 |
| Drinking water | 3 | **1.09 (1.00,1.18)** | 0.0 | 0.884 |
| *Testing method* | | | | |
| HG-AFS | 2 | **1.34 (1.01,1.76)** | 0.0 | 0.557 |
| ICP-MS | 4 | 1.01 (0.97,1.05) | 65.9 | 0.032 |
| *As Exposure level* | | | | |
| 0–1 μg/l | 2 | 0.99 (0.94,1.05) | 36.3 | 0.210 |
| 1–10 μg/l | 5 | **1.13 (1.06,1.21)** | 0.0 | 0.499 |
| 10–100 μg/l | 1 | 1.00 (0.99,1.01) | 0.0 | <0.000 |
| **LBW (As)** | | | | |
| *Region* | | | | |
| Africa | 1 | 1.09 (0.91,1.31) | 0.0 | <0.000 |
| America | 2 | **1.07 (1.01,1.13)** | 0.0 | 0.653 |
| Asia | 2 | 1.01 (0.89,1.14) | 12.1 | 0.286 |
| Europe | 1 | 0.52 (0.10,2.83) | 0.0 | <0.000 |
| *Sample* | | | | |
| Blood | 1 | 1.30 (0.79,2.14) | 0.0 | <0.000 |
| Urine | 2 | 0.99 (0.96,1.03) | 6.6 | 0.301 |
| Drinking water | 2 | **1.07 (1.01,1.13)** | 0.0 | 0.653 |
| *Testing method* | | | | |

*(Continued)*

**Table 2.** (Continued)

| Subgroup | Number of literature | Pooled estimates (95%CI) | I² (100%) | P |
|---|---|---|---|---|
| HG-AFS | 2 | 1.19 (0.70,2.02) | 3.4 | 0.309 |
| ICP-MS | 2 | 0.99 (0.96,1.03) | 6.6 | 0.301 |
| *As Exposure level* | | | | |
| 0–1 μg/l | 4 | **1.07 (1.02,1.13)** | 0.0 | 0.842 |
| 1–10 μg/l | 0 | / | / | / |
| 10–100 μg/l | 1 | 0.99 (0.99,0.99) | 0.0 | <0.000 |

Note: (/): no value; Data in bold indicate significant results; As, arsenic; SGA, small for gestational age; LBW, low birth weight; PTB, preterm birth; HG-AFS,Hydride generation atomic fluorescence spectrometry; ICP-MS,Inductively-coupled plasma mass spectrometry.

significantly associated with an increased risk of PTB (OR: 1.13, 95% CI: 1.06, 1.21). However, no significant statistical differences were observed in other regions and sample types (Table 2).

**Association between as exposure and LBW.** The meta-analysis results indicated no significant association between As exposure and the occurrence of LBW (Fig 3). The overall analysis showed an odds ratio (OR: 1.03, 95% CI: 0.97, 1.09). However, to identify potential sources of heterogeneity (I² ≥ 50%), a subgroup analysis was conducted. The subgroup analysis results based on region, sample type, detection method, and arsenic exposure levels were presented in Table 2. In two studies conducted in the United States, a correlation between As exposure and LBW was found (OR: 1.07, 95% CI: 1.01, 1.13). In four studies involving As exposure levels of 0–1 μg/L, low-level maternal As exposure was associated with LBW (OR: 1.13, 95% CI: 1.06, 1.21).

## Meta-regression analysis results

The meta-regression analysis results showed that, in the analysis of SGA and PTB, the regression coefficients for most variables did not reach statistical significance, indicating that these variables had a weak or non-significant impact on the outcomes. However, in the analysis of LBW, the Region variable had a statistically significant effect on LBW (P = 0.047), suggesting that considering regional factors may be particularly important when assessing the impact of As exposure on IUGR risk. Additionally, for certain variables (e.g., Sample and Testing method) in explaining the heterogeneity in PTB and LBW, the adjusted R² showed negative values, suggesting that these models may not effectively explain the variation between studies. A summary of other meta-regression analysis results can be found in Table 3 and is illustrated in Fig 1–12 of S1 Fig.

## Sensitivity analysis results

The sensitivity analysis indicated that the exclusion of most individual studies had minimal impact on the overall effect estimates in the meta-analysis, confirming the robustness of the results. However, after excluding the study by Kyi Mar Wai (2017) [35], the effect estimate changed significantly, suggesting that the combined results may not be stable. Therefore, these results should be interpreted with particular caution (S2 Fig).

## Assessment of publication bias

Publication bias in the included studies was assessed using funnel plots, the Egger test, and the Begg test. The funnel plots indicated some degree of publication bias across studies on SGA,

**Table 3. Meta-regression of As exposure and IUGR risk.**

| Variables | Number of obs | Tau² | I² (100%) | Adjusted R-squared(100%) | Coefficient (95%CI) | Std. Err. | *P-value* |
|---|---|---|---|---|---|---|---|
| **SGA (As)** | | | | | | | |
| Region | 8 | 0.00 | 57.62 | 100.00 | 0.18 (-0.07,0.45) | 0.11 | 0.131 |
| Sample | 6 | 0.03 | 84.24 | -32.85 | -0.01(-0.32,0.31) | 0.11 | 0.966 |
| Testing method | 6 | 0.03 | 76.39 | -17.70 | -0.07(-0.71,0.56) | 0.23 | 0.765 |
| As Exposure level | 7 | 0.03 | 75.61 | -40.42 | -0.02(-0.32,0.28) | 0.12 | 0.870 |
| **PTB (As)** | | | | | | | |
| Region | 9 | 0.00 | 54.96 | -60.57 | -0.01(-0.20,0.19) | 0.08 | 0.963 |
| Sample | 6 | 0.01 | 50.47 | 4.40 | -0.07(-0.27,0.12) | 0.07 | 0.356 |
| Testing method | 6 | 0 | 56.19 | 100.00 | -0.30(-0.88,0.30) | 0.21 | 0.246 |
| As Exposure level | 8 | 0.01 | 63.82 | -73.01 | 0.03(-0.10,0.154) | 0.05 | 0.630 |
| **LBW (As)** | | | | | | | |
| Region | 6 | 0 | 0.00 | 100.00 | -0.07(-0.13,0.00) | 0.02 | **0.047** |
| Sample | 5 | 0 | 0.00 | 100.00 | -0.08(-0.16,0.01) | 0.03 | 0.066 |
| Testing method | 4 | 0.00 | 5.03 | 77.05 | -0.20(-1.26,0.87) | 0.25 | 0.514 |
| As Exposure level | 5 | 0 | 0.00 | 100.00 | -0.08(-0.16,0.01) | 0.03 | 0.060 |

Note: Data in bold indicates significant results; As, arsenic; SGA, small for gestational age; LBW, low birth weight; PTB, preterm birth.

PTB, and LBW outcomes. Although the Egger test revealed publication bias in the association between As exposure and SGA (P < 0.05), no significant bias was observed in the other outcomes (P > 0.05, see S3 Fig).

## Discussion

In this study, the association between As exposure and IUGR (SGA, PTB, and LBW) was evaluated through a systematic review and meta-analysis. The results indicated a significant positive correlation between As exposure and SGA, suggesting that As may be an important environmental risk factor affecting fetal development. Subgroup analysis further revealed regional differences in this association, particularly in Asia, where As exposure was significantly associated with an increased risk of SGA. Although the overall associations between As exposure and PTB and LBW were weaker, significant associations were observed in certain specific subgroups (e.g., studies using blood samples, specific testing methods, or higher exposure levels). Additionally, our meta-regression analysis explored other potential sources of heterogeneity, highlighting that regional factors may be particularly important when assessing the impact of As exposure on IUGR risk. These findings underscore the importance of reducing As exposure during pregnancy to lower the risk of IUGR.

As is a toxic metal widely distributed in the environment through both natural and anthropogenic activities. Humans are primarily exposed to As through contaminated air, water, and food [43,44]. Of particular concern is the accumulation of As metabolites in body tissues, especially in the placenta, which can have significant negative impact on normal fetal development. The effect of prenatal As exposure on fetal development, particularly its association with the risk of IUGR [45], has garnered considerable attention. Multiple studies have explored the accumulation of As in the body and its effects on fetal development. Research indicates that As exposure may lead to abnormally elevated levels of insulin-like growth factor 1 (IGF-1), which is closely associated with the occurrence of IUGR. Additionally, the accumulation of As metabolites in the placenta may trigger oxidative stress and inflammatory responses, impairing placental function, and affecting fetal nutrient supply, thereby leading to

IUGR [46]. Further studies suggest that As may affect fetal development through epigenetic mechanisms. Specifically, As may alter DNA methylation and histone modifications, resulting in abnormal gene expression that affects fetal organ development, particularly the development of vital organs. This may manifest as IUGR at birth and also increase the risk of disease in offspring early in life [47].

SGA refers to fetal growth restriction where the birth weight is below the 10th percentile for gestational age [48]. The occurrence of SGA is often associated with fetal malnutrition, poor maternal health, and exposure to environmental toxins [49]. Existing epidemiological studies and animal experiments provide substantial evidence that environmental As exposure may lead to fetal growth restriction and the occurrence of SGA by affecting placental function [22]. The results of this meta-analysis show a significant association between As exposure and SGA, with a combined odds ratio (OR: 1.06, 95% CI: 1.00, 1.13), suggesting that As exposure may increase the risk of SGA. Related studies also indicate that As exposure may negatively impact fetal growth by disrupting placental function, inducing oxidative stress, and triggering inflammatory responses [50]. Subgroup analysis revealed regional differences in the risk of SGA associated with As exposure. Specifically, in Asia, As exposure was significantly associated with an increased risk of SGA (OR: 1.28, 95% CI: 1.10, 1.49) [36,37], which may be related to higher levels of environmental As pollution and As content in drinking water in this region. Interestingly, these findings are consistent with findings from studies in Bangladesh [51]. In contrast, studies conducted in the United States and Europe did not observe a significant association, possibly reflecting lower levels of As exposure or more effective control measures in these regions. Additionally, a significant positive correlation was found between SGA and As levels in blood (OR: 1.39, 95% CI: 1.14, 1.70) [37,39] as well as As levels in urine (OR: 1.35, 95% CI: 1.06, 1.73) [32,36]. Further analysis of exposure levels revealed that maternal exposure to higher levels of As (≥74 µg/L) significantly increased the risk of SGA (OR: 1.25, 95% CI: 1.04, 1.50), while lower levels of As exposure did not show a significant effect. This suggests that As may have a threshold effect, where adverse effects on fetal development only become apparent at certain exposure levels. Moreover, sensitivity analysis and the Begg test further validated the stability of the results. However, the Egger test indicated the potential presence of publication bias in the association between As exposure and SGA ($P < 0.05$), suggesting that more well-designed studies are needed in the future to verify the association between As exposure and the risk of SGA. Meta-regression analysis did not find significant associations between country, sample type, detection method, and As exposure levels with SGA. This suggests that these variables may not be the primary sources of heterogeneity in SGA, or the existing data may be insufficient to detect these relationships.

In recent years, increasing attention has been given to the association between As exposure and the risk of PTB. This growing concern stems from the widespread presence of As exposure among pregnant women and its potential health risks. PTB, defined as birth before 37 weeks of gestation, is commonly associated with maternal health conditions, malnutrition, and exposure to environmental toxins [12]. The meta-analysis found that although there was no relationship between As exposure and the risk of PTB, in specific subgroups, an increased risk of PTB was associated with As exposure. Specifically, subgroup analysis revealed a significant positive correlation between As exposure and PTB risk in the "Africa" subgroup (OR: 1.20, 95% CI: 1.05, 1.37), with a more pronounced association observed under certain sample types and detection methods. For instance, studies using blood samples for detection showed a significant association between As exposure and PTB (OR: 1.35, 95% CI: 1.02, 1.78). Similarly, studies using drinking water as the sample also observed a significant association (OR: 1.09, 95% CI: 1.00, 1.18) [33,34]. Moreover, studies employing the HG-AFS technique demonstrated a significant association between As exposure and PTB (OR: 1.34, 95% CI: 1.01, 1.76)

[37,38], supporting the positive correlation between As exposure and PTB risk. Furthermore, subgroup analysis further revealed a dose-dependent relationship between As exposure and PTB. The studies found that maternal exposure to 1–10 μg/L As was significantly associated with an increased risk of PTB (OR: 1.13, 95% CI: 1.06, 1.21) [33,34,37,40], which is consistent with studies conducted in the United States [52], indicating that even low levels of As exposure may increase the risk of PTB. To sum up, our findings suggest that future research should focus on the impact of sample type, detection methods, and varying As exposure levels on the risk of PTB.

LBW refers to infants born with a birth weight of less than 2500 grams and is typically associated with pregnancy complications, maternal malnutrition, lifestyle factors, and exposure to environmental toxins [53]. LBW is a risk factor for infant morbidity and mortality and may have long-term health implications, including an increased risk of metabolic diseases, cardiovascular diseases, and cognitive developmental disorders [54]. Evidence suggests that prenatal As exposure may interfere with normal fetal growth through various mechanisms, leading to the occurrence of LBW [55]. However, the meta-analysis showed that the association between As exposure and LBW was not statistically significant (OR: 1.03, 95% CI: 0.97, 1.09). Despite this, related studies have shown a significant association between prenatal As exposure and reduced birth weight [56]. Additionally, subgroup analysis and meta-regression analysis provided more specific research evidence. For example, in two studies conducted in the United States, a significant association between As exposure and LBW was found (OR: 1.07, 95% CI: 1.01, 1.13), with both studies using drinking water as the test sample. Notably, the meta-regression analysis also indicated that the region variable had a statistically significant effect on LBW (P = 0.047). This result suggests that regional factors may be particularly important when assessing the impact of As exposure on fetal development risk. Furthermore, in studies involving low-level As exposure (0–1 μg/L), low-dose As exposure significantly increased the risk of LBW (OR: 1.13, 95% CI: 1.06, 1.21) [34,37,40]. These findings suggest a stronger association between As exposure and increased LBW risk in certain regions and under specific As exposure levels, which is consistent with previous studies [57]. Therefore, future research should place greater emphasis on the influence of regional factors and As exposure levels on the association between As exposure and LBW risk.

This study has the following strengths: 1. This meta-analysis on the association between As exposure and IUGR addresses the inconsistencies in previous studies and provides comprehensive evidence. 2. The study conducted a thorough assessment of the risk of bias in the included studies using multiple bias risk assessment tools. 3. Subgroup analysis and meta-regression analysis were used to explore sources of heterogeneity, offering a basis for better interpretation of the results and application of the evidence. However, this study also has several limitations: 1. Most of the included studies were cohort studies (10 out of 11), making it impossible to conduct subgroup analysis based on study design to explore sources of heterogeneity. 2. High heterogeneity was observed in the analysis of As exposure and SGA, which may be due to differences in As exposure measurement methods, exposure levels, and sample types across studies. Although subgroup analysis was conducted to explore sources of heterogeneity, sensitivity analysis indicated that the stability of the results should be interpreted with caution. 3. The samples were primarily from the Americas, Europe, Asia, and Africa. While these regions are covered, they may not fully represent the global population. More studies from different countries and regions are needed to validate the association between As exposure and IUGR. 4. Although most studies used advanced techniques (such as ICP-MS and HG-AFS) to measure As exposure, differences in biological samples and exposure levels across studies may introduce potential bias. 5. Some studies had limitations in handling missing data and reporting secondary outcomes. 6. The assessment of publication

bias indicated that some studies might have publication bias, which should be interpreted with caution. More well-designed studies are needed in the future to validate our findings.

In conclusion, the findings of this study are consistent with previous epidemiological research and further support the association between As exposure and an increased risk of IUGR. Given the widespread nature of As exposure and its potential threat to maternal and fetal health, especially in areas with severe environmental As pollution, it is crucial to implement stricter public health policies and environmental regulations to reduce As exposure during pregnancy and protect maternal and infant health. However, due to the observed heterogeneity and potential publication bias, these results should be interpreted with caution.

## Conclusions

This study suggests that As exposure may significantly increase the risk of IUGR, particularly SGA, with the association being more pronounced in certain regions and under specific exposure conditions. Although the overall association between As and PTB and LBW is weaker, significant correlations were still observed in certain subgroups. Therefore, more diverse studies are needed in the future to further validate and explore these associations, supporting the development of relevant public health policies.

## Supporting information

**S1. Statements.**
(DOCX)

**S1 Table. PRISMA 2020 checklist.**
(DOCX)

**S2 Table. Search strategy.**
(DOCX)

**S3 Table. NTP/OHAT risk of bias rating tool.**
(DOCX)

**S4 Table. Summary of studies identified in literature search.**
(DOCX)

**S5 Table. Study screening and exclusion details.**
(DOCX)

**S6 Table. Included references for meta-analysis.**
(DOCX)

**S7 Table. Data extraction table.** Note: As, arsenic; SGA, small for gestational age; LBW, low birth weight; VLBW, very low birth weight; PTB, preterm birth; VPTB, very preterm birth; USA, United States of America; CO, cohort study; CS, cross-sectional study.
(DOCX)

**S8 Table. Quality assessment results of included studies.**
(PDF)

**S1 Fig. Meta-regression analysis results.**
(PDF)

**S2 Fig. Results of sensitivity analyses.**
(PDF)

**S3 Fig. The results of the bias assessment were published.**
(PDF)

## Author contributions

**Conceptualization:** jing Jiang, Xuan Zuo, Hong Pan, Yan Xie.

**Data curation:** jing Jiang, Xuan Zuo.

**Formal analysis:** jing Jiang, Xuan Zuo, Songlin An, jing Yang, Qiongdan Hu, Lu Fan.

**Funding acquisition:** jing Jiang, jing Yang, yuanzhong Zhou, Yan Xie.

**Investigation:** Linfei Wu, Rong Zeng, Qiongdan Hu, Lu Fan, Haiyu Wang, Chuanwu Yang, Yihan Liang.

**Methodology:** yuanzhong Zhou, Hong Pan, Yan Xie.

**Software:** Rong Zeng.

**Validation:** jing Jiang, Songlin An, jing Yang.

**Visualization:** Linfei Wu, Haiyu Wang.

**Writing – original draft:** jing Jiang, Xuan Zuo, Songlin An.

**Writing – review & editing:** yuanzhong Zhou, Hong Pan, Yan Xie.

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
