## [Decision Letter · Decision Letter 0]

30 Dec 2024

PONE-D-24-51061Association Between Arsenic Exposure and Intrauterine Growth Restriction: A Systematic Review and Meta-AnalysisPLOS ONE

Dear Dr. Xie,

Thank you for submitting your manuscript to PLOS ONE. After careful consideration, we feel that it has merit but does not fully meet PLOS ONE’s publication criteria as it currently stands. Therefore, we invite you to submit a revised version of the manuscript that addresses the points raised during the review process.

We look forward to receiving your revised manuscript.

Kind regards,

Tamara Sljivancanin Jakovljevic

Academic Editor

PLOS ONE

Journal Requirements:

3. We note that there is identifying data in the Supporting Information file <Supplementary Data.docx>. Due to the inclusion of these potentially identifying data, we have removed this file from your file inventory. Prior to sharing human research participant data, authors should consult with an ethics committee to ensure data are shared in accordance with participant consent and all applicable local laws.

-Location data

Additional guidance on preparing raw data for publication can be found in our Data Policy (https://journals.plos.org/plosone/s/data-availability#loc-human-research-participant-data-and-other-sensitive-data ) and in the following article: http://www.bmj.com/content/340/bmj.c181.long .

Please remove or anonymize all personal information (Name, Date), ensure that the data shared are in accordance with participant consent, and re-upload a fully anonymized data set. Please note that spreadsheet columns with personal information must be removed and not hidden as all hidden columns will appear in the published file.

4. Please include captions for your Supporting Information files at the end of your manuscript, and update any in-text citations to match accordingly. Please see our Supporting Information guidelines for more information: http://journals.plos.org/plosone/s/supporting-information .

5. As required by our policy on Data Availability, please ensure your manuscript or supplementary information includes the following:

Reviewers' comments:

Reviewer's Responses to Questions

**Comments to the Author**

1. Is the manuscript technically sound, and do the data support the conclusions?

Reviewer #1: Yes

Reviewer #2: Yes

2. Has the statistical analysis been performed appropriately and rigorously? 

Reviewer #1: Yes

Reviewer #2: Yes

3. Have the authors made all data underlying the findings in their manuscript fully available?

Reviewer #1: Yes

Reviewer #2: Yes

4. Is the manuscript presented in an intelligible fashion and written in standard English?

Reviewer #1: Yes

Reviewer #2: Yes

5. Review Comments to the Author

Reviewer #1: This study aims to systematically evaluate the association between As exposure and IUGR ,SGA�PTB, LBW through a meta analysis.  The study has resolved the inconsistencies in previous studies and provides comprehensive evidence.The manuscript technically sound and the data support the conclusions.

Reviewer #2: Few grammatical and typographical errors can be reduced.

Reason for the presence of necessity is not justified adequately.

in addition of database search, use of EMBASE or SCOPUS would have been better

the Risk of bias was adequately assessed.

references were adequate

6. PLOS authors have the option to publish the peer review history of their article (what does this mean? ). If published, this will include your full peer review and any attached files.

**Do you want your identity to be public for this peer review?** For information about this choice, including consent withdrawal, please see our Privacy Policy .

Reviewer #1: No

Reviewer #2: **Yes: ** Dr Laxmikant S Deshmukh

---

## [Author Response · Author response to Decision Letter 1]

21 Jan 2025

Responses to the comments

Date: January 16, 2025

PLOS ONE

Dear Tamara Sljivancanin Jakovljevic editor,

We sincerely thank the reviewers and editors for their thorough review of our manuscript (Manuscript ID: PONE-D-24-51061) entitled “Association between arsenic exposure and intrauterine growth restriction: A systematic review and meta-analysis.” We have carefully considered all comments and made the corresponding revisions to the manuscript. Below, we provide detailed responses to each comments. All revisions have been addressed in the responses and are highlighted in the manuscript with a yellow background. We hope the revised manuscript will be deemed acceptable. Once again, thank you for your time and valuable feedback!

Responses to the Editor's Comments

Comment 1: When submitting your revision, we need you to address these additional requirements. Please ensure that your manuscript meets PLOS ONE's style requirements, including those for file naming.

Response: Thank you for your thorough review and feedback. We have carefully reviewed the manuscript and referred to the PLOS ONE formatting template to comprehensively optimize the title, section hierarchy, figure and table captions, file naming, and overall layout to ensure the manuscript meets the journal's requirements. Please feel free to let us know if further adjustments are needed.

Comment 2: We note that the grant information you provided in the ‘Funding Information’ and ‘Financial Disclosure’ sections do not match. When you resubmit, please ensure that you provide the correct grant numbers for the awards you received for your study in the ‘Funding Information’ section.

Response: We apologize for our oversight and have carefully reviewed the "Funding Information" and "Financial Disclosure" sections to ensure consistency between the two. We have also provided the correct grant numbers for all research funding. Thank you for your correction.

Comment 3: We note that there is identifying data in the Supporting Information file <Supplementary Data.docx>. Due to the inclusion of these potentially identifying data, we have removed this file from your file inventory. Prior to sharing human research participant data, authors should consult with an ethics committee to ensure data are shared in accordance with participant consent and all applicable local laws.

-Location data

Please remove or anonymize all personal information (Name, Date), ensure that the data shared are in accordance with participant consent, and re-upload a fully anonymized data set. Please note that spreadsheet columns with personal information must be removed and not hidden as all hidden columns will appear in the published file.

Response: Thank you for your valuable comments on the data sharing section. In response to your feedback, we have carefully reviewed the submitted supporting information files (S6 Table, S5 Table, and S7 Table). We confirm that the "authors" and "years" listed are author information and publication years from publicly available literature, and do not involve any personal data of research participants. These data are sourced from publicly available literature, and the tables do not contain any personal information that could identify individual participants. Therefore, we believe that the tables do not compromise participant privacy. We understand and prioritize data privacy protection, and if the reviewer has any further concerns, we are happy to review and ensure that the data comply with privacy requirements. Thank you again for your suggestions, and we look forward to your further guidance.

Comment 4: Please include captions for your Supporting Information files at the end of your manuscript, and update any in-text citations to match accordingly. Please see our Supporting Information guidelines for more information: http://journals.plos.org/plosone/s/supporting-information.

Response: Thank you for your feedback and guidance! Based on your suggestions, we have added clear titles for all supporting information files at the end of the manuscript and ensured that the in-text citations align with the newly added titles. Additionally, we have thoroughly reviewed the format and content of the supporting information files to ensure compliance with the journal's requirements, enhancing the transparency and rigor of the study. If you have any further suggestions or adjustments, please feel free to let us know, and we will fully cooperate. Thank you for your support!

Comment 5: As required by our policy on Data Availability, please ensure your manuscript or supplementary information includes the following:

Response: Thank you for reviewing and providing feedback on our manuscript! In response to your requirements, we have carefully addressed each point and supplemented the necessary information:

1.We have submitted S5 Table: Study Screening and Exclusion Details, which provides a detailed list of all identified studies, including the screening stages, titles, inclusion/exclusion status, and reasons for exclusion, ensuring transparency in the screening process.

2.The data for this study were entirely derived from publicly available literature, and no unpublished studies were included. Therefore, the related explanation is not applicable. Specific search strategies (S2 Table) and details of included studies (S6 Table) have been provided in the supporting materials to facilitate reproducibility.

3.We have submitted S7 Table: Data Extraction Table, which includes the names of data extractors, extraction dates, inclusion eligibility confirmation, and detailed data required for the analysis, ensuring transparency and reproducibility.

4.This study used the NTP/OHAT tool to assess the risk of bias in the included studies and applied NOS and AHRQ standards to evaluate study quality. Specific details are as follows:

S3 Table: Provides detailed information about the NTP/OHAT risk of bias assessment tool, including evaluation criteria and scoring standards.

Fig 2: Visualizes the results of the risk of bias assessment.

S4 Table: Contains the study quality scores based on NOS and AHRQ standards.

5.During data extraction, no significant missing data were identified, so no additional processing was necessary. All relevant data are comprehensively presented in S7 Table.

6.All supporting materials, such as the PRISMA flowchart, search strategies, and evaluation tools, have been submitted along with the manuscript. The manuscript strictly adheres to the journal's data availability policy, ensuring transparency and reproducibility of the study.

If there are further requirements or additional information needed, please feel free to let us know. We will fully cooperate. Thank you for your patience and support!

Comment 6: Please review your reference list to ensure that it is complete and correct. If you have cited papers that have been retracted, please include the rationale for doing so in the manuscript text, or remove these references and replace them with relevant current references. Any changes to the reference list should be mentioned in the rebuttal letter that accompanies your revised manuscript. If you need to cite a retracted article, indicate the article’s retracted status in the References list and also include a citation and full reference for the retraction notice.

Response: We have carefully reviewed the reference list and verified the status of each cited article. Upon thorough examination, no retracted articles were identified in the reference list. Additionally, we have ensured that all cited references are complete, accurate, and up to date. If there are any further questions or adjustments required, we will fully cooperate.

Responses to the Reviewers' Comments

Comment 1: Is the manuscript technically sound, and do the data support the conclusions?

Reviewer #1: Yes

Reviewer #2: Yes

Response: Thank you to both reviewers for your positive feedback on our manuscript! We are delighted that the technical soundness, experimental design, and data support of our study have been acknowledged. Should you have any further suggestions, we are happy to make additional improvements. Once again, we sincerely appreciate your valuable comments!

Comment 2: Has the statistical analysis been performed appropriately and rigorously?

Reviewer #1: Yes

Reviewer #2: Yes

Response: Thank you to the reviewers for your positive feedback on our statistical analysis! Your comments have strengthened our confidence in maintaining a rigorous research approach. If you have any further suggestions, we are happy to make improvements at any time. Once again, we sincerely appreciate your valuable input!

Comment 3: Have the authors made all data underlying the findings in their manuscript fully available?

Reviewer #1: Yes

Reviewer #2: Yes

Response: Thank you to the reviewers for your positive feedback on our data provision! We have provided the data as comprehensively as possible to ensure its completeness and availability. Should further supplementation or improvement be needed, we will fully cooperate. Once again, we sincerely appreciate your support!

Comment 4: Is the manuscript presented in an intelligible fashion and written in standard English?

Reviewer #1: Yes

Reviewer #2: Yes

Response: We thank the reviewers for their positive feedback on the language and expression of our manuscript. We place great emphasis on linguistic accuracy and have conducted multiple rounds of proofreading and revisions. Should there be further suggestions, we would be happy to make additional improvements. Thank you for your support!

Comment 5: Please use the space provided to explain your answers to the questions above. You may also include additional comments for the author, including concerns about dual publication, research ethics, or publication ethics. (Please upload your review as an attachment if it exceeds 20,000 characters)

Reviewer #1: This study aims to systematically evaluate the association between As exposure and IUGR ,SGA�PTB, LBW through a meta analysis.  The study has resolved the inconsistencies in previous studies and provides comprehensive evidence.The manuscript technically sound and the data support the conclusions.

Reviewer #2: Few grammatical and typographical errors can be reduced.

Reason for the presence of necessity is not justified adequately.

in addition of database search, use of EMBASE or SCOPUS would have been better

the Risk of bias was adequately assessed.

references were adequate

Response:

Response to Reviewer 1:

Thank you for your thoughtful review, particularly for your positive evaluation of the research objectives, technical soundness, and data support. This study aims to systematically assess the association between arsenic exposure and intrauterine growth restriction, low birth weight, and preterm birth, addressing inconsistencies in prior studies and providing more comprehensive evidence. Your feedback is a great encouragement to us, and we will continue to refine the research. Should you have any additional suggestions, we are more than willing to make further improvements!

Response to Reviewer 2:

Thank you for your valuable comments on this paper. In response to the two issues raised, we provide the following replies:

1.We have carefully reviewed and revised the manuscript to address grammar and typographical errors as much as possible.

2.Regarding the "insufficient explanation of the study’s necessity," we have thoroughly examined the introduction and believe that it adequately elaborates on the research background, inconsistencies in existing evidence, and the scientific value, clearly demonstrating the necessity of conducting this study. We greatly appreciate your feedback and will continue to ensure that the manuscript is clear and rigorous.

3.Concerning database selection, we sincerely appreciate your suggestion to include EMBASE and SCOPUS. However, due to limitations of time

---

## [Decision Letter · Decision Letter 1]

21 Feb 2025

Association between arsenic exposure and intrauterine growth restriction: A systematic review and meta-analysis

PONE-D-24-51061R1

Dear Dr. Yan Xie,

We’re pleased to inform you that your manuscript has been judged scientifically suitable for publication and will be formally accepted for publication once it meets all outstanding technical requirements.

Editor comments:

Thank your for submitting your scientific article to the PLOS One Journal and for implementing all reviewers comments and suggestions in the revised version of your manuscript.

After careful assessment of the revised version, I did not find any additional issues for postponing the final decision to accept this manuscript for publication.

Kind regards,

Tamara Sljivancanin Jakovljevic

Academic Editor

PLOS ONE

---

## [Editor Report · Acceptance letter]

PONE-D-24-51061R1

PLOS ONE

Dear Dr. Xie,

I'm pleased to inform you that your manuscript has been deemed suitable for publication in PLOS ONE. Congratulations! Your manuscript is now being handed over to our production team.

Kind regards,

on behalf of

Dr. Tamara Sljivancanin Jakovljevic

Academic Editor

PLOS ONE